# A Method to Construct an Indoor Air Pollution Monitoring System Based on a Wireless Sensor Network

**DOI:** 10.3390/s19040967

**Published:** 2019-02-25

**Authors:** Ahras Naziha, Li Fu, Galloua Mohamed Elamine, Lingling Wang

**Affiliations:** 1School of Automation Science and Electrical Engineering, Beihang University, Beijing 100191, China; naz-aero@hotmail.com (A.N.); fuli@buaa.edu.cn (L.F.); 2School of Computer Science and Engineering, Beihang University, Beijing 100191, China; galloua.ma@hotmail.fr

**Keywords:** wireless sensor network, region of interest, voronoi diagram, point in polygon, coverage, coverage hole

## Abstract

Wireless Sensors Networks (WSNs) are currently receiving much research interest due to their wide-ranging use is a number of different fields. In the current study, a system based on a WSN is proposed that can monitor indoor air pollution in several public spaces, such as subway stations, offices, schools, and hospitals. The proposed system uses integrated sensors in mobile phones, moving from a stationary nodes model to a mobile nodes model. The main objective of building this system is to provide full coverage of the target area. To achieve this goal, the system is simulated by MATLAB and the following algorithms are applied: Particle Swarm Optimization (PSO) to maximize the coverage in the region of interest (RoI), Voronoi Diagram (VD) to detect holes in the coverage, and finally the Point in Polygon (PiP) algorithm to heal the holes in the coverage. The application of the algorithms mentioned above has been very effective as PSO has increased the coverage rate of the monitoring area to 100%. The VD allowed us to define the exact location of coverage holes whilew the Point in Polygon algorithm allowed us to heal the holes and find the remaining sensors in order to improve network coverage. This enabled us to achieve full coverage of the monitoring area.

## 1. Introduction 

With the rapid development of communication technology, network and remote sensing technologies can help us to improve air pollution monitoring systems that are usually designed in a wireless mode. Currently, monitoring the physical infrastructure of data centers is very important so as to enable effective maintenance, the prevention of any downtime, accurate event information, as well as efficiently planning appropriate energy requirements. These technologies are usually designed in a wireless mode and create a lot of air pollution. Furthermore, physical electronic devices that are used alongside complex electronic devices create a lot of air pollution within a specific frequency spectrum [1,2,3,4]. Wireless air pollution monitoring systems operate in several modes However, these modes are expensive to install and maintain due to their complexity. Although WSNs is initially used for military and industrial reasons [5], it has been developed for a growing number of industrial applications in recent years. 

A study from 2009 [6] presented a network for indoor and outdoor air quality monitoring by installing nodes in different rooms that included tin dioxide sensor arrays connected to an acquisition and control system. The main findings of this paper were: The development of an air quality monitoring system that uses a smart sensor in a wireless network where the nodes have a specific location, the embedding of neural network processing blocks distributing the processing charge between the embedded systems (web sensor) and the web browser installed in a personal computer, the development of PC software for sensing node TCP/IP remote control, advanced data processing, data storage, and web publishing software associated with air quality monitoring systems using JavaScript and LabView for a web publisher. The work in reference [7] presented a prototype WSN that could monitor the environment in a conference room in a smart building using stationary sending nodes. This prototype network facilitated the testing of four different architectures tasked with learning the relationhip between CO_2_ and occupant state. Four different learning models were compared: Gradient Boosting (GB), k-Nearest Neighbors (KNN), Linear Discriminant Analysis (LDA), and Random Forests (RF). The KNN model returned the best performance under preferred settings with an accuracy of 47.8% and RMSE (Root Mean Square Error) of 1.021.

The system in reference [8] aimed to detect the level of seven gases in three different settings, ozone (O_3_), particulate matter, carbon monoxide (CO), nitrogen oxides (NO_2_), sulfur dioxide (SO_2_), volatile organic compound, and carbon dioxide (CO_2_), on a real-time basis. In doing so, the system provided an overall air quality alert using one of two sensor nodes. In the sensing node, a smoothing algorithm was introduced to prevent temporary sensor errors, and an aggregation algorithm was used to reduce network traffic and power consumption. In the middleware, all data from the sensor cloud were calibrated with the characteristic tables of the cross sensitivities and temperature/humidity dependence. Experiments were conducted to validate and support the development of the system for real-time monitoring and alerts.

The authors in [9] addressed the concept of a real-time WSN capable of monitoring the concentration of indoor carbon dioxide and dangerous situations inside three different areas (two medium size and one by size) using stationary sensor nodes for data aggregation and mobile nodes for data calibration. They followed a cognitive networking technique with an opportunistic routing protocol. An information processing framework was proposed to detect outliers, form data packets, and calibrate the sensors. The system managed to report CO_2_ concentration in real-time, while the information processing framework minimized outliers. In [10], indoor air quality was improved by controlling particulate matter (PM) consisting of components such as house dust and bacteria, using an atmospheric pressure plasma. In this study, two types of electrode were used for particle collection and formaldehyde removal. Both electrodes were designed to shorten the gap distance and to minimize the discharge voltage used for indoor air treatment. Formaldehyde control was performed in a 1 m^3^ chamber. A high gas flow was used with the electrode system to simulate the large flow rate in a room. These electrode systems had lower operating voltages than those reported for conventional ESPs.

In the field of technology, there are many indoor air pollution monitoring systems based on WSN. The network of those systems was built with stationary sensor nodes and developed using different architectures, modules, networking protocols, and expensive elements such as: IoT platform and Libelium collections [11], AVR ATmega64 micro controller, Raspberry, and LoRaWAN wireless communication [12], XBee module and Arduino Mega system [13], and ZigBee networking protocol [14,15], GSM modem [16], and ARIMA prediction models [17]. However, the data collected, and the analyzed results, are not accessible to the general public.

For those major reasons, in this project we will realize a low-cost system to monitor indoor air pollution and switch from stationary nodes in a specific area to mobile nodes in the whole city. We will achieve this by using the mobile phones of the general public to compose a WSN and then transmit the values of the parameters collected by using WIFI or a hot spot to the base station where the level of air pollution will be evaluated and then communicated to the general public. Deploying sensor nodes is a critical phase that dramatically affects the performance of the network and the quality of service. Additionally, the performance of the WSN is highly influenced by the method used to deploy the sensor nodes.

A number of different issues have been raised in regard to the deployment of sensor nodes in a WSN. Several studies based on optimization algorithms have considered different methods to achieve total coverage of the monitoring area such as: Biogeography based optimization and deferential evaluation [18], virtual forces algorithm [19], genetic algorithm with particle swarm optimization [20], particle swarm optimization with Voronoi Diagram (VD) [21], and differentiated deployment algorithm based on image processing and 3D modeling [22]. These studies mainly concern stationary and mobile cases.

The main objective of this project is to ensure the best quality of service, and this begins with full coverage of the monitoring area. The process of deploying the sensors has an impact on the performance of the WSN because it affects the network coverage, communication cost, and management resources, so the deployment strategy of a WSN is a major problem.

After we had conducted our review of the relevant literature, we then began to choose the algorithms that would suit us in our work to allow us overcome the aforementioned problems. Therefore, for the first part of coverage optimization and deployment strategy, the Particle Swarm Optimization (PSO) will be applied to fulfill the coverage over the region of interest (RoI) in Section 2. The second part of our application will be divided into two sections: A VD to detect the location of the coverage holes holes, and to then use the Point in Polygon (PiP) for coverage hole healing, which will be expanded in Section 3. Finally, we will use MATLAB language to simulate our system in Section 4, and give the conclusion in Section 5.

## 2. WSN Coverage Optimization Based on PSO

### 2.1. Network Node Set Coverage 

Assume that the target area (Region of Interest) A is a two-dimensional plane, which is digitized into l×w grids where the size of each grid is equal to 1. The WSN with n total number of sensors is deployed on the target area with two specifications:
(1).All sensors have the same sensing range r and communication range $R_{c}$ = 2 r(2).The sensor nodes with the same parameters are placed, and the coordinates of each node are ($x_{i},y_{i}$), where  i=1,2, …,n.

The monitoring field of the i sensor node is centered on sensor position coordinate ($x_{i},y_{i}$) and r is the circle of the monitoring radius. The node of each sensor node set is denoted by $\;C=\left\{c_{1},c_{2},\ldots ,c_{n}\right\}$, where $\;c_{i}=\left\{x_{i},y_{i},r\right\}$. Assuming that the monitoring area A is digitally discredited into l×w pixels where the coordinates of each pixel are denoted by ($x_{j},y_{j}$) and  j=1,2, …, l×w. The Euclidean distance [23] between the j pixel and the i  sensor node is: (1)$$d\left(c_{i},p_{i}\right)=\sqrt{{\left(x_{j}-x_{i}\right)}^{2}+{\left(y_{j}-y_{i}\right)}^{2}}$$

The probability of the event which the pixel ($x_{j},y_{j}$) is covered by the sensor node $\;c_{i}=\left\{x_{i},y_{i},r\right\}$ is: (2)$$P_{cov}\left(x_{j},y_{j},\;c_{i}\right)=\{\begin{array}{c} 1,\;if\;d\left(c_{i},p_{i}\right)<r\;\\ 0,\;otherwise\; \end{array}$$

Due to the interference of environment, noise, and other factors in sensing model, the sensor nodes should distribute in a certain probability [24],
(3)$$P_{cov}\left(x_{j},y_{j},\;c_{i}\right)=\;\{\begin{array}{c} 1,\;if\;d\left(c_{i},p_{i}\right)\leq r-re\\ e^{\frac{-\alpha_{1}\lambda_{1}\beta_{1}}{\lambda_{2}\beta_{2}+\alpha_{2}},\;if\;r-re<d\left(c_{i},p_{i}\right)<r+re\;}\\ 0,\;otherwise\; \end{array}$$
where re is the reliability parameter of the measuring of sensor nodes with re(0<re<r), $\alpha_{1},\beta_{1},\alpha_{2},\beta_{2}>0$ are the measurement parameters about the characteristics of the sensor node, where 

(4)$$\lambda_{1}=re-r+\;d\left(c_{i},p_{i}\right)\;\mathrm{and}\;\lambda_{2}=re+r-d\left(c_{i},p_{i}\right)$$

Considering that the target pixel is simultaneously covered by multiple sensor nodes [24], the joint monitoring probability is
(5)$$P_{cov}\left(Cov_{j}\right)=1-\underset{c_{i}\in Cov_{j}}{\prod }(1-P_{cov}\left(x_{j},y_{j},\;c_{i}\right))\;i=1,2,\ldots ,n;\;j=1,2,\;\ldots ,\;l\times w$$
where $Cov_{j\;}$ is the set of sensor nodes that measure the target pixel  j.

### 2.2. Optimization Model of WSN Coverage

In this work, we use the joint monitoring probability of the node set to measure whether each pixel of the target area is covered. $P_{th}$ is the expected coverage threshold, then: (6)$$P_{cov}\left(Cov_{j}\right)=\;\{\begin{array}{c} 1,\;if\;P_{cov}\left(Cov_{j}\right)<\;P_{th}\\ 0,\;if\;P_{cov}\left(Cov_{j}\right)\;\geq P_{th}\; \end{array}$$
where $P_{cov}\left(Cov_{j}\right)=1$ indicates that the target pixel j is overwritten; otherwise, the target pixel j is not overwritten [23]. 

The effective coverage points are calculated by the previous formula, and the number of sampling points of the whole target area is l×w, so the coverage rate ϕ of the monitoring area A is calculated with the following formula: (7)$$\phi =\frac{\sum_{j}P_{cov}\left(Cov_{j}\right)}{l\times w}$$

The objective function in this work is the monitoring area coverage rate ϕ, the node in the monitoring area establishes the wireless sensor network coverage optimization model according to the previous formula.
(8)$$maximize\;\phi =\frac{\sum_{j}P_{cov}\left(Cov_{j}\right)}{l\times w}$$
where $\{\begin{array}{c} 0\leq x_{i\;}\leq l\\ 0\leq y_{i}\leq w \end{array}$.

### 2.3. Network Coverage Optimization Based on PSO 

The PSO is inscribed in the family of evolutionary algorithms and is one of the most popular meta-heuristic optimization algorithms inspired by nature. This technique is used for any problem to explore a search in space to find the set parameters that maximize/minimize a particular goal [25]. 

The population in the PSO algorithm is called the swarm, and each individual in the group is called the particle. The displacement of any particle (as indicated above) is governed by very specific rules and conditions and is influenced by the movement of other particles in the neighborhood.

The algorithm is guided by personal experience (Pbest), overall experience (Gbest), and the present movement of the particles to decide their next positions in the search space. In D dimensional search space, we have M particles at certain speed flight. To find the optimal solution, PSO algorithm initializes a group of random particles through the iterative. With the speeds of each generation of particles, the location updates the objective function as follows [26],
(9)$$v_{ij}\left(n+1\right)=w\times \;v_{ij}\left(n\right)+c_{1}\times rand_{1}\times \left(pbest_{ij}-x_{ij}\right)+c_{2}\times rand_{2}\times \left(gbest-x_{ij}\right)$$
j: Refers to the dimension of the question searched by particlesi: Refers to the number of particlesn: Refers to iterations$c_{1}\;and\;c_{2}:$ The accelerating constants in the algorithm$rand_{1},\;rand_{2}:$ Are two independent uniformly distributed random variables between (0,1)$v_{ij}$: Is usually valued in a certain rangew: Is the inertia coefficient where:
(10)$$w=wmax-\frac{n}{\max number}\;\times \left(wmax-wmin\right)$$

The appreciation of the quality of its position is stopped by the value of the “objective” function at this point. It is essential that this particle can memorize the better position by which it has already passed, formulated as follows:
$$Pbest_{ij}=\left(\;pbest_{i1},\;pbest_{i2},\;\ldots ,\;pbest_{iD}\right)$$ The best position reached by the particles of the swarm is formulated as follows:
$$Gbest=\left(gbest_{1},\;gbest_{2},\;\ldots ,\;gbest_{D}\right)$$

The notion of Gbest (global best), is modeled on the PSO Global version (PSOG) where all the particles of the swarm are derived from the particle  i. At the beginning of the algorithm, the particles of the swarm are initialized randomly/regularly in the search space. Subsequently at each iteration, the particles move, merging the three components:
$v_{ij}\left(n\right)$: Is the speed of the particle in last iterative;$c_{1}\times rand_{1}\times \left(pbest_{ij}-x_{ij}\right)$: The “cognitive” part which is the distance between the current position with best position of the particle, which indicates that particles can learn on itself;$c_{2}\times rand_{2}\times \left(gbest-x_{ij}\right)$: The “social” part which is the distance between the current position with the global the best position, that represents the collaboration between the particles.

At the end of the displacement of the particles in a given iteration, the new positions are evaluated and the two vectors Pbesti and Gbest are reindexed. According to Equation (11) in the case of a minimization of an objective function then Equation (12) in the PSOG can be achieved: (11)$$Pbest_{i}\left(t+1\right)=\{\begin{array}{c} Pbest_{i}\left(t\right),\;si\;f\left(x_{i}\left(t+1\right)\right)\geq Pbest_{i}\left(t\right)\\ x_{i}\left(t+1\right),\;otherwise \end{array}$$
(12)$$Gbest\left(t+1\right)=\arg\min_{Pbesti}f\left(Pbest_{i}\left(t+1\right)\right),\;1\leq i\leq N\;$$

This procedure is represented in the PSO algorithm, for N represents the number of particles of the swarm: 

Step 1: Randomly initialize N particles: Position and speed.

Step 2: Evaluate particle positions.

Step 3: For each particle *i*, $Pbest_{i}=x_{i}$

Step 4: Calculate Gbest according to (13)

Step 5: As if the stopping criterion is not satisfied, move the particles according to (9) and (10), assess particle positions and update $Pbest_{i}$ and Gbest according to (12) and (13).

## 3. Coverage Hole Detection and Healing

The previous phase of work allowed us to identify the positions of the sensors used to build the air pollution monitoring wireless network and ensure full coverage of the RoI. After a period of time (one hour), it is necessary to ascertain if any of the sensors change its position or leave the RoI. 

Updating the sensor locations allow to check the positions of the deployed sensors. If they remain their positions we keep the same deployed sensor for RoI coverage otherwise we move to the next phase of the work which is the holes detection and healing process. The second phase of project will be divided into two parts: The first is holes detection and coverage based on the VD, and the next is to repair the holes using the PiP algorithm.

### 3.1. Coverage Hole Detection Based on the VD

As the sensors used in this work are embedded in public mobile phones, so these nodes move randomly, which is why the network is able to crack at any time (mobile phone is turned off, leaves the RoI, is disconnected, etc.). To define the coverage hole introduced by the off-grid sensor and its exact location, we will introduce the VD algorithm to provide each sensor with their own cell. In this way we can define with accuracy the coverage hole location.

A VD is a partition of a space into small regions called the polygon of Voronoi based on closeness to points in a specific subset of the plane as each region contains one, and only one, site (point) and each point belongs to its region (cell) [27]. In this application, the VD devised the RoI into n cells where each sensor is the center of its own cell. To detect the coverage hole, we use the following theorems shown in Figure 1.

### 3.2. Coverage Hole Healing Based on PiP Algorithm 

After defining the location of the cell that contains a coverage hole using the VD, the next step is to find other sensors inside this cell in order to reapply the coverage algorithm and select the suitable sensors to deliver full coverage in the RoI. To complete this stage, we will use the PiP algorithm.

The PiP algorithm allows us to check, by way of a program, whether a particular point is inside a polygon or outside of it. A common way to approach the problem is to count how many times a line drawn from the point (in any direction) crosses the edge of the polygon. If the line and the polygon intersect an even number of times (or not at all), the point is outside. If they cross an odd number of times, the point is inside. This is true even for complex shapes that have a lot of coordinates and a very precise edge [28]. This process can be seen in Figure 2.

After having selected the sensors which are inside the cell of the coverage hole, we will apply to these sensors the algorithm of coverage already applied in the previous phase of this project (from function 3 to 8).

## 4. Simulation and Results Discussion

In this study, assuming that target area (RoI) A is two-dimensional plane (100 m^2^), it is digitized into 100×100  grids where each grid size is equal to 1. In the simulation, the WSN with 100 total numbers of sensors is deployed on the target area with two assuming specifications:
(1).All sensors have the same sensing range r = 20 m and communication range $r_{c}$ = 2 r(2).Each sensor knows its location by a certain mechanism and the location information can be sent to the base station.

This project is defined in order to find the best distribution of the deployed sensors to obtain full coverage of the monitoring area. To achieve our objective, the system implementation will be passing through three basic phases:Coverage optimization of the monitoring area using PSO algorithm to find an optimal number of the deployed sensors to totally cover the RoI.Coverage hole detection based on the VD.Coverage hole healing using the algorithm of PiP to find sensors inside the hole then to use them to revalue the coverage of the hole using the coverage algorithm used in the first phase of work.

The effectiveness of the network coverage optimization strategy will be determined through a simulation experiment. Using a computer with a 1.10 GHZ processor, the coverage optimization algorithm of the WSN is carried out in MATLAB (R2013a) environment.

### 4.1. WSN Coverage Optimization 

#### 4.1.1. Application Method

In the simulated two-dimensional target area, assuming that all the deployed sensors in a big space have a large sensing range *r* = 20 m, and each sensor knows its location by a certain mechanism and sends it to the base station. The coverage optimization based on the PSO algorithm should calculate the coverage rate of each pixel node for each sensor node according to Equation (4), calculate the joint coverage of each pixel node for each sensor node according to Equation (5), and calculate the coverage rate of the RoI using the objective function according to Equation (8), in the particle swarm algorithm also known as fitness.

In this work, we assume that there are M particles (the total number of the sensors in RoI, M = 100). The condition for the simulation experiment is described in the Table 1. The PSO algorithm is as follows: Randomly generate the position and velocity of each particle in the monitoring area; Update the position and velocity of each particle according to Equations (9–11); Calculate the fitness of each particle according to the optimization objective function; Compare the particle fitness and the fitness of its own best position, if better, and then set the new pbestCompare the fitness of each particle and the fitness of the best position in the population, if better, set the new gbestUntil the maximum number of iterations is reached, the algorithm stops, otherwise proceed to step twoOutput the optimal fitness and the corresponding particle position.

#### 4.1.2. Results and Discussion

In the simulation condition mentioned above, the detection radius *r* will be constant and equal to 20 m, and the number of sensors used will vary between 20 and 40 to determine its impact on the performance of the coverage. The iterations-coverage rate and the coverage effect are shown in Table 2, and Figure 3, Figure 4, Figure 5 and Figure 6.

During the optimization of coverage by using the PSO algorithm, the coverage optimization went through four stages, first the deployment of 20 sensors, then 30 sensors, followed by 35 sensors, and finally 40 sensors. The simulations results are as follows: The 20-sensor deployment result graph started with 84% coverage, after that the graph started to increase until it reached 87% at iteration 230 then jumped directly to the optimal result 88% and stabilized.The 30-sensor deployment result graph started with a 92% coverage rate and reached almost 95% after few iterations and remained practically stable until 150 iterations. After that, the algorithm started looking for a better result until iteration 225 where the final optimal coverage rate was 96%.The 35-sensor deployment result graph started with a 95% coverage rate, and reached almost 97% just after few iterations. After that, the algorithm started looking for a better result until iteration 280, where the final optimal coverage rate was 98%.The 40-sensor deployment result graph started with 97% coverage and increased to 99% at iteration 120 and maintained that rate until iteration 200. After that, the algorithm started looking for a better result. In this application the monitoring area was totally covered, and the rate reached 100%.

According to the results presented above, the curves of the coverage rate, with and without optimization according to the number of the deployed sensors, are shown in Figure 7.

The analysis of the Figure 7 shows us that the PSO algorithm was more efficient when the number of sensors used was 20 where the percentage of the coverage rate had increased from 70% to 88%. This was revised in principle of the PSO algorithm which attempted to find a better solution in bad conditions.

It is also noted that the curve slope between the 20 sensors and 35 sensors was large compared to the curve slope between the 35 sensors and 40 sensors. We can conclude that in a RoI with our characteristics (L×W,r,n,m), the optimal number of deployed sensors to have total coverage of the monitoring area varied between 35 and 40 sensors.

In reference [23], 20 sensors from total 40 sensors were used to completely cover an area of 20 m^2^. Therefore, the number of deployed sensors was optimized by 50%. In our work, the deployment of 40 sensors from a total number of 100 sensors was enough to fully cover an area of 100 m^2^. Therefore, we optimized our deployment by 60%. It is obvious that the PSO algorithm successfully found the best deployment strategy for this application. The position and the number of deployed sensors were determined to fulfill the goal of maximum coverage of the monitoring area.

### 4.2. Coverage Holes Detection

#### 4.2.1. Application Method

The PSO algorithm allowed us to optimize a total coverage of the RoI and to collect the best positions of the deployed sensors. After a period of time (one hour), the base station updated these positions leading to one of the deployed sensors out of the coverage network. Following this, the VD will be used to define the position of the coverage hole.

The target area (RoI) A is also a two-dimensional plane (100 m^2^). It is digitized into 100×100  grids where each grid size is equal to 1. The total number of deployed sensors in coverage optimization is 40 sensors. In this process, we will first randomly move one of these sensors and assume that, after updating the sensor positions, it has been found that the sensor has pulled out of the coverage network as shown in Figure 8. Following this, the 39 sensors remaining in the RoI (their positions are already known) are used to partition the RoI into 39 cells and to discover the location of the coverage hole using the VD assuming that the theorems 1 and 2 are already defined.

#### 4.2.2. Results and Discussion

Using the VD algorithm, the monitoring area was divided into several cells using the selected sensors for full coverage. Each sensor had its own cell that gave us a 40-cell area with no coverage hole found. 

The sensor indicated in Figure 8 was randomly chosen to exit the coverage network. The VD algorithm was reapplied to the remaining 39 sensors in the area to find the coverage holes introduced by the outgoing sensor. The result is shown in the Figure 9.

It was assumed that, after a period of time (one hour), the base station updated the positions of the 40 sensors deployed and that one of the sensors came out of the network. Therefore, the next application started by randomly selecting a sensor to exit the network and then to apply the VD algorithm on the remaining 39 sensors in the network to discover the location of the coverage holes. The results of this application are shown in Figure 9 and Figure 10. 

According to Figure 10a, it is noted that the sensor (13) that left the network was at the neighborhood of sensors (34), (25) and (28) so they then covered the space already covered by the sensor (13) but their sensing range was not enough to cover all the space. For this reason, we found a coverage hole between cell (27) and (33), as shown in Figure 10b.

### 4.3. Coverage Holes Healing

The next step in this application was to search for the sensors that were originally in the coverage hole cells (cell 27 and 33) by applying the PiP algorithm. Following this, we reapplied the coverage code to find the best sensor to enhance the coverage. After applying the PiP algorithm in the coverage hole cells, we found three sensors in this area as shown Figure 11.

After finding five sensors between cells 27 and 33, we then reapplied them to the coverage algorithm to ascertain the best sensor that could cover the hole and join it to the monitoring area network. The result is shown in Figure 12.

To heal the coverage hole found by the VD algorithm, the PiP algorithm was applied to find the sensors inside the hole cells. The PiP algorithm found three sensors in these two cells and then the coverage algorithm was reapplied to select one of these three sensors. We then latter selected a sensor that covered the hole and increased the coverage rate to 99.13%.

## 5. Conclusions

A system that monitors indoor air pollution by combining wireless sensing technology, PSO, VD, and PiP algorithms is proposed in this paper. A PSO algorithm successfully found the best deployment strategy for this application. The simulation results indicate that in a RoI with our characteristics (l×w, r, n, m), the optimal number of deployed sensors varies between 35 and 40 sensors to have total coverage of the monitoring area. Because the initial positions of the sensors were randomly selected, the results will not be the same each time. Admittedly, there may be a small possibility that the initial deployment plan is the optimal one. However, the objective is unchanged and fulfilled by adopting the PSO algorithm, which is to find the best deployment strategy of the sensor network. Moreover, VD successfully finds the exact location of the hole coverage which allowed the application of PiP algorithm inside those holes and found a new join sensor to enhance the network. However, PSO is a heuristic algorithm that requires a big server to get fast results. Therefore, to apply this work in real life in the future, the base station must use big servers for a good quality of service.

## Figures and Tables

**Figure 1 sensors-19-00967-f001:**
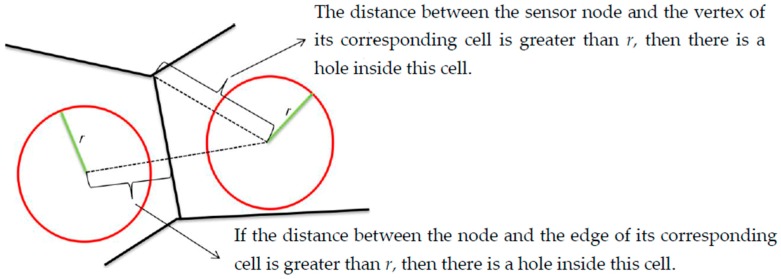
Coverage holes detection method.

**Figure 2 sensors-19-00967-f002:**
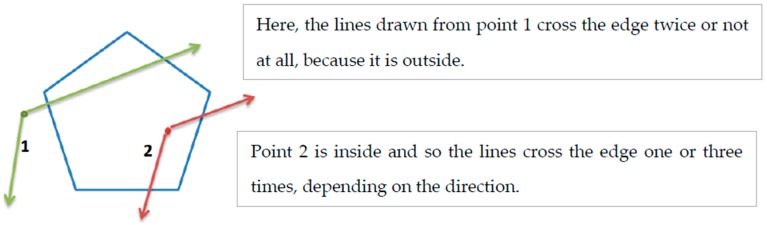
The principle of the point in polygon algorithm.

**Figure 3 sensors-19-00967-f003:**
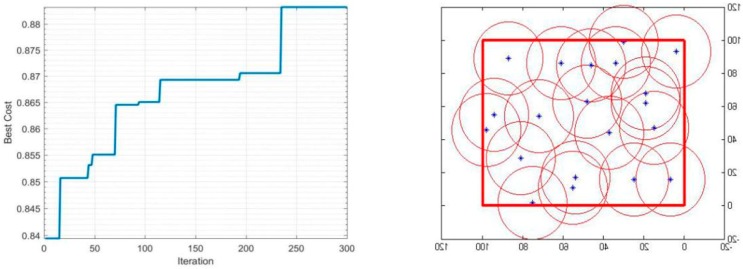
Coverage of 20 sensors.

**Figure 4 sensors-19-00967-f004:**
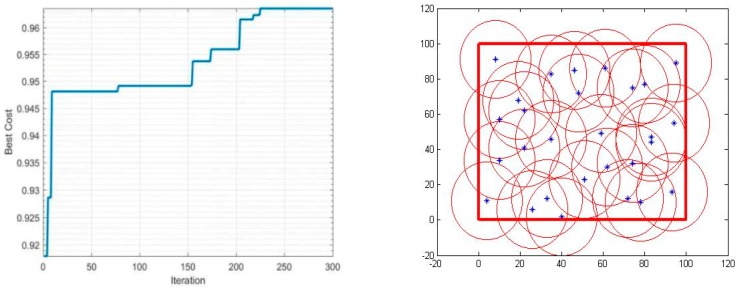
Coverage of 30 sensors.

**Figure 5 sensors-19-00967-f005:**
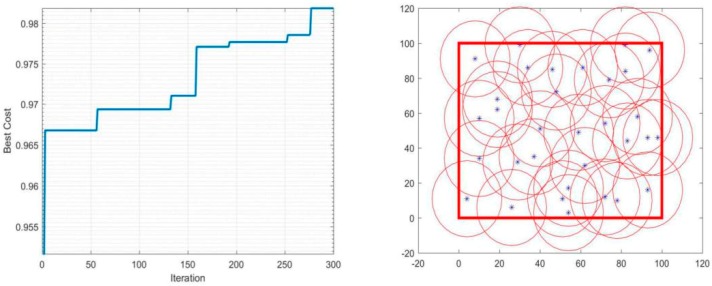
Coverage of 35 sensors.

**Figure 6 sensors-19-00967-f006:**
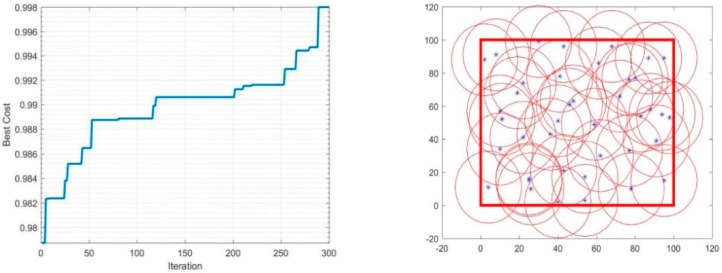
Coverage of 40 sensors.

**Figure 7 sensors-19-00967-f007:**
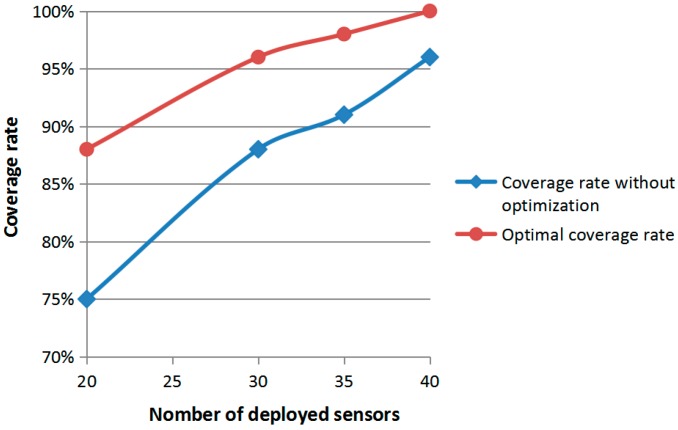
Coverage rate compared to the number of sensor deployment.

**Figure 8 sensors-19-00967-f008:**
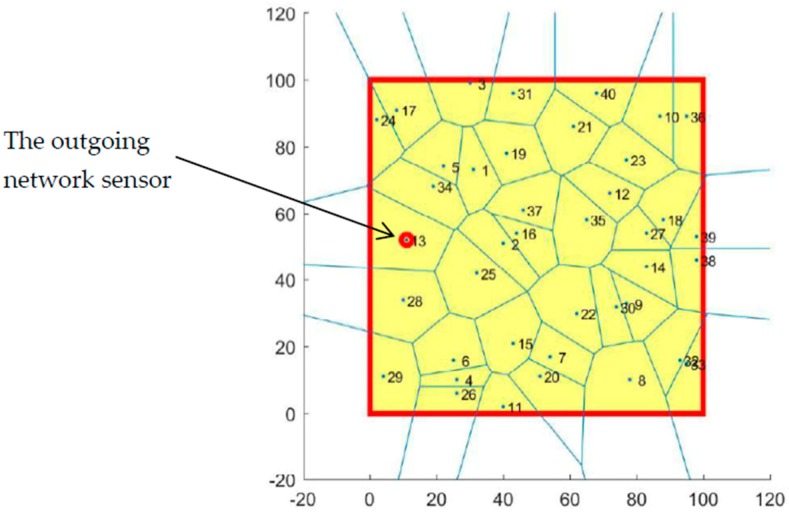
Voronoi cells of the 40 deployed sensors.

**Figure 9 sensors-19-00967-f009:**
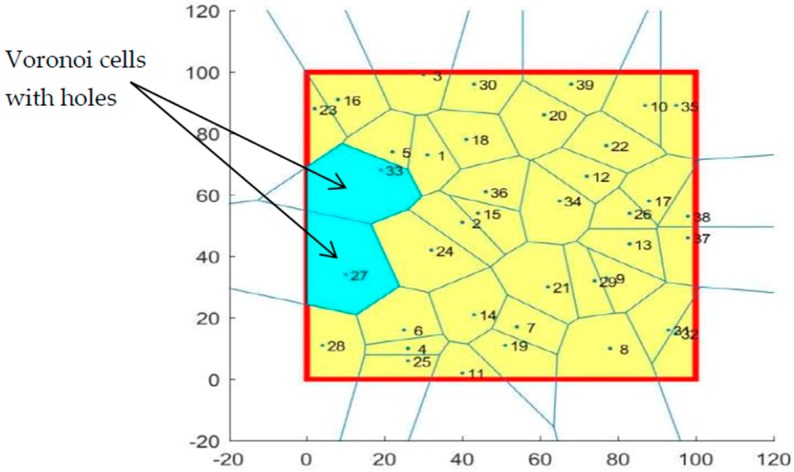
Voronoi cells of remaining 39 sensors.

**Figure 10 sensors-19-00967-f010:**
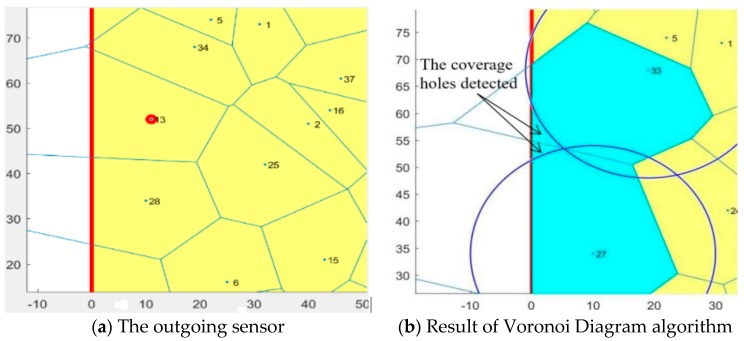
The coverage hole detected by the Voronoi Diagram algorithm.

**Figure 11 sensors-19-00967-f011:**
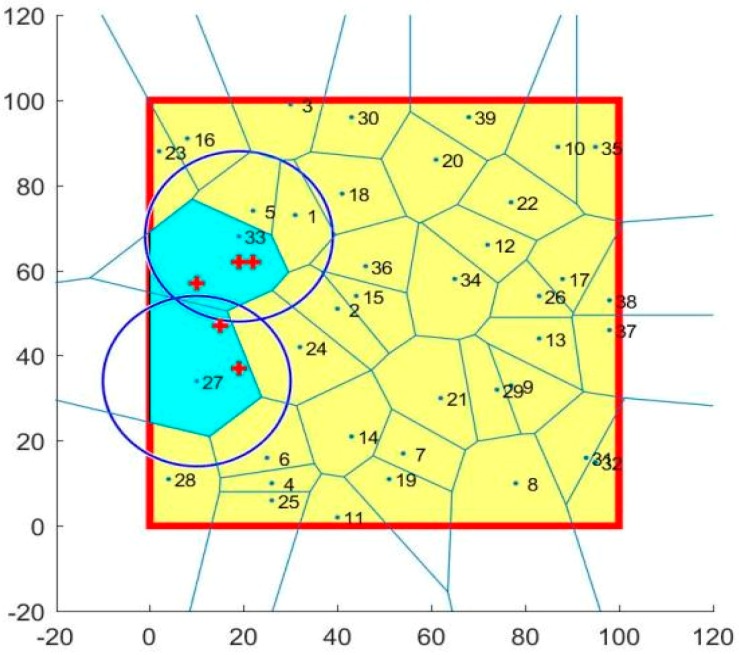
Point in polygon results.

**Figure 12 sensors-19-00967-f012:**
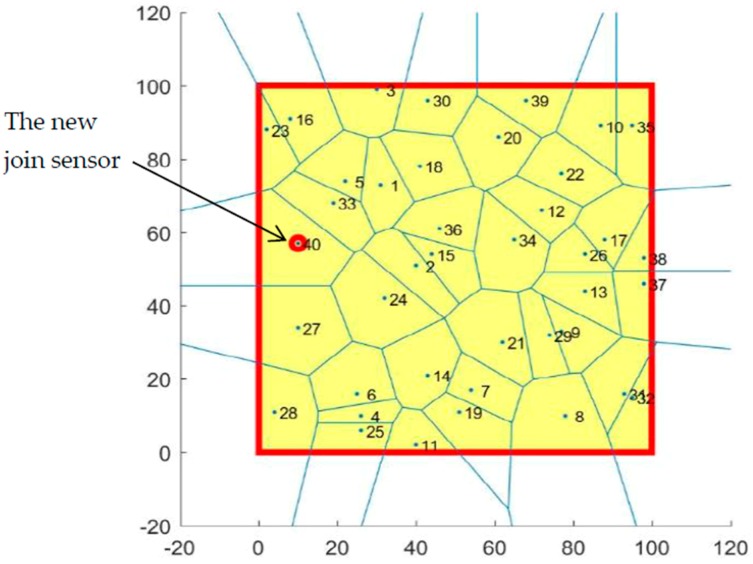
The distribution of the new sensors in the monitoring area network.

**Table 1 sensors-19-00967-t001:** The condition for simulation experiment.

Population M	Deployed Sensors Number N	Swarm Size D=2×N	$c_{1},c_{2}$	$\alpha_{1},\alpha_{2}$ $\beta_{1},\beta_{2}$	$re=\frac{r}{2}$	Maximum Iterations
100	20	40	2	1,0,1, 0.5	10	300
100	30	60	2	1,0,1, 0.5	10	300
100	35	70	2	1,0,1, 0.5	10	300
100	40	80	2	1,0,1, 0.5	10	300

**Table 2 sensors-19-00967-t002:** Coverage optimization results.

Area A network coverage optimization performance test				
Sensor node number N	20	30	35	40
Coverage rate without optimization	75%	88%	91%	96%
Optimal coverage (%)	88%	96%	98%	100%

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
