# Peer review of "A Method to Construct an Indoor Air Pollution Monitoring System Based on a Wireless Sensor Network"

_sensors, 2019, doi:10.3390/s19040967_

Reviewer 1 Report

The paper presents an interesting approach for indoor air pollution monitoring with WSNs, along with simulation results. Some of the following suggestions could improve the paper.

- The simulation description should be improved. Why the range is set in 20 m? There are a few prototypes, such as the following:

Prototypes of Opportunistic Wireless Sensor Networks Supporting Indoor Air Quality Monitoring

which uses 20 m for transmission range, due to the sensor constraints. Is this the same case here?

- If there are changes in the simulation parameters, how will these changes affect the results? If the transmission range is smaller, what would happen with monitoring area coverage of the proposed solution?

-There are a few grammar mistakes that should be improved.

- Which simulator was used for the experiments? Also, what is the associated energy cost per node? When the number of nodes increases the performance is better, however the energy cost and the total cost increases as well.

- The related work is weak. Consider including the following journal papers and compare your work with them

Smart Sensors Network for Air Quality Monitoring Applications

Designing learned CO2-based occupancy estimation in smart buildings

ISSAQ: An Integrated Sensing Systems for Real-Time Indoor Air Quality Monitoring

Real-Time Indoor Carbon Dioxide Monitoring Through Cognitive Wireless Sensor Networks

Basic Study of Indoor Air Quality Improvement by Atmospheric Plasma

Reviewer 2 Report

Comments to the authors:

I suggest same minor corrections.

General:

-          Equation number lining?

-          There are two Figures 11??

-          There are two Tables 1???

-          Where is in the tex ref 18 ???

Page 1: 1. Introduction:

-line 29: The author should insert: “Nowadays event monitoring in Physical infrastructure of Data Centers are very important because of effective maintenance, action preventing any downtime, accurate event information as well as appropriate energy support. This technology is usually designed in wireless mode and make great air pollution. On the other hand, the electronic physical devices produce electromagnetic interferences between different complex electronic devices, and make great air pollution in specific frequency spectrum, ref (A-D).” Insert ref:

A

-          Matko Vojko, Brezovec Barbara. Improved data center energy efficiency and availability with multilayer node event processing. Energies, ISSN 1996-1073, 2018, 11, 9, 1-17, doi: 10.3390/en11092478.

B

-          Jia, M.; Srinivasan, R.S.; Raheem, A.A. From occupancy to occupant behavior: An analytical survey of data acquisition technologies, modeling methodologies and simulation coupling mechanisms for building energy efficiency. Renew. Sustain. Energy Rev. 2017, 68, 525–540.

C

-          Podberšič Marko, Matko Vojko, Šegula Matjaž. An EMI filter selection method based on spectrum of digital periodic signal. Sensors, 2006, vol. 6, iss. 3, p. 90-99. https://www.mdpi.com/1424-8220/6/3/90/htm

D

-          Antonini, G. SPICE Equivalent Circuits of Frequency – Domain Responses. IEEE Transactions on Electromagnetic Compatibility 2003, 45 (3), 502-512. https://ieeexplore.ieee.org/document/1223619

-           

Conclusion:

What are disadvatnages of the proposed method?

Author Response

The authors would like to thanks the reviewers for their careful review and constructive comments, and the editor for handling the efficient review process. We have followed the reviewers’ suggestions, and the revised the paper accordingly. This summarizes the changes we have made in the revision to address the concerns raised by the reviewers. The reviewers’ comments are cited in italic front and our responses follow in normal front.

1.  Genaral:

I modified the number lining in the paper.

There are two sections in figure 11: (a) and (b)

There is just one table in Table 1 and is modified

The reference [18] is added in the text

2.  Introduction:

Thanks for your valuable suggestion. I have added the paragraph that you have proposed in the first section of the introduction and inserted the references. 

3. Conclusion:What are disadvatnages of the proposed method?

We used in our proposed method PSO to optimize the deployment of the sensors to full cover the target area and which is a heuristic algorithm.  The implementation was in computer with 1.10 GHZ processor which takes more than 10 hours to take results. So to apply this work in real life, the base station must use big server for good quality of service.

Round  2

Reviewer 1 Report

The authors followed most of the recommended changes. Regarding the 20m transmission range, the authors could site the suggested paper as a reference of a real prototype that uses 20m transmission range for air pollution monitoring. 

Author Response

The authors would like to thank the reviewer for the careful review and constructive comments, and the editor for handling the efficient review process. We have followed the reviewer’s suggestions, and revised the paper accordingly. This note summarizes the changes we have made in the revision to address the concerns raised by the reviewers. 

First, we have checked the paper and modified the spelling mistakes highlighted in blue in the paper.

Second, according to the reviews valuable comment, we give the explanation about 20m transmission range in section 4.1.1. In the simulated two-dimensional target area with two assuming specifications, we assume that all the deployed sensors in a  big space have a large sensing range r=20m and each sensor knows its location by a certain mechanism and sends it to the base station. Then the PSO, Voronoi Diagram (VD) and Point in Polygon (PiP) algorithms for wireless sensor network are verified.